# Nature of Science (NOS) Being Acquainted with Science of Science (SoS): Providing a Panoramic Picture of Sciences to Embody NOS for Pre-Service Teachers

**Ismo T. Koponen**

Department of Physics, University of Helsinki, 00014 Helsinki, Finland; ismo.koponen@helsinki.fi;
Tel.: +358-2941-50652

**Abstract:** Understanding about nature of science is important topic in science education as well as in pre-service science teacher education. In science education, Nature of Science (NOS), in its different forms of educational scaffoldings, seeks to provide with students an understanding of features of scientific knowledge and science in general, how scientific knowledge changes and becomes accepted, and what factors guide scientific activities. For a science teacher, deep and broad enough picture of sciences is therefore of importance. This study attempts to show that the research field called Science of Science (SoS) can significantly support building such a panoramic picture of sciences, and through that, significantly support NOS. The SoS approaches the structure and dynamics of science quantitatively, using scientific documents (e.g., publications, reports, books and monographs and patent applications) as trails to map the landscape of sciences. It is argued here that SoS may provide material and interesting cases for NOS, and in so doing enrich NOS in a similarly significant way as history, philosophy and sociology of science (HPSS) scholarship has done thus far. This study introduces several themes based on SoS that are of relevance for NOS as they were introduced and discussed in a pre-service science teachers' course. The feedback from pre-service teachers shows that introducing SoS, with minimal additional philosophical interpretations and discussions, but simply as evidential facts and findings, sparks ideas and views that come very close to NOS themes and topics. Discussions related to nature of science, and specific educational NOS scaffoldings for it, can find a good companion in SoS; the latter providing facts and evidence of thee structure and dynamics of sciences, the former providing perspectives for interpretations.

**Keywords:** nature of science; science of science; science; scientific knowledge; scientometrics



## 1. Introduction

Nature of Science (NOS) is today an integral part of science education as well as a recommended part of science teacher education [1–4]. Its purpose is to provide students with a picture of science and scientific knowledge that is simple enough to be useful and viable for purposes of school education, but that would still convey a sufficiently authentic view of science and present it as a part of larger societal endeavours [2,3,5–8]. The importance of NOS as part of a school curriculum, as well as its themes, are broadly agreed, which has resulted in shared core, which is referred to as a consensus view of NOS [1]. However, if there is a consensus of basic themes, the debate on how NOS should approach and reach its goals are continuing [9–15]. Many surveys have also shown that scientists have views and opinions about science that are not in concordance with the views contained in NOS [10,11,16–19]. Debates surrounding NOS and its assumed faultlines have been discussed and summarised in several reviews see, e.g., [3,9,13,19–26]), and some alternatives to the original versions of NOS have been suggested [22,23,27]. Here, such discussions are not repeated, nor alternative versions of NOS are suggested to correct or straighten the supposed fault lines. However, to put the present study in a perspective

of wider discussion about NOS, a summary is made of some of the fault lines that have tangential contact with topics discussed here.

The consensus view of NOS has been criticised for being too vague and insensitive to important disciplinary differences, and in addition, providing a false picture of science, of the nature of its knowledge as well as of the practices of science (see, e.g., [9,11–16,19,21,25,26,28]). Some authors have claimed that NOS is essentialist in the sense that it seeks the essential nature of science, and dogmatic in that it seeks integrated and consolidated views [20]. In large part, such discussions of NOS repeat many themes familiar from early criticism that NOS does not correspond to the views of science philosophers (see, e.g., [29–31]), or those of practicing scientists [17,18]. The former criticism was addressed by Schwartz et al. [32,33], and the latter by Abd-El-Khalick [1]. In both cases, it was noted that consensus NOS is quite open to different positions and allows great flexibility regarding how ideas and views at different levels of sophistication can be accommodated within its general schemes. According to Abd-El-Khalick [1], the assumed essentialism or dogmatism are largely consequences of reading too much into simplifications that are meant to make NOS appropriate for school-level instruction, but are easily relaxed at higher levels of education where more in-depth discussions are appropriate [1]. Abd-el-Khalick [1], in providing arguments supporting consensus NOS and its broadness of views, points out that consensus NOS distances it from too detailed epistemological and philosophical commitments, and that this distancing can be seen rather an advantage than a shortcoming.

In regard requirements that NOS should better take into account the views of practicing scientists, one encounters the problem that these views are very context-dependent and have very little coherence with regard to epistemology (see, e.g., [16]). As such, they provide awkward grounding for developing practical solutions for teaching, although they would otherwise provide insights on how and why scientist differ in their views about epistemological questions.

The basic goals that NOS seeks to fulfill are broad and general: to provide an understanding of features of scientific knowledge and science in general, how scientific knowledge changes and becomes accepted, and what kinds of factors guide scientific activities. Such goals are of particular importance to pre-service science teacher education. In practice, however, it is not uncommon to find that the differing views of scientists and science teacher educators create certain tensions similar to those documented in research focusing on scientists' views about NOS themes. However, it is not necessary to change NOS so that it could accommodate the different views of practicing scientists, or on the other hand, better correspond to certain preferred philosophical stances like some version of realism. On the contrary, we can see the strength of NOS in its ambivalence to such positions, which allows its use as a basic scheme to see science through different lenses and provide a scaffolding to recognise what kinds of lenses one uses.

The practical purpose of this study is to discuss how a recent research field called Science of Science [34] can contribute to NOS. It is argued that many findings and notions of Science of Science are relevant to NOS, and that SoS can provide material for NOS to discuss different aspects and features of science from perspectives a chosen NOS viewpoint offers. The viewpoint adopted on NOS in this study comes close to consensus NOS in acknowledging the necessity of flexibility and broadness taking distance from specific philosophical epistemological positions. The categorization of themes as contained in consensus NOS is found here appropriate for discussing the topic of the present study. For practical purposes of organizing the presentation, this study adopts a similar kind of thematic categorization as in consensus NOS. This, however, does not mean endorsing or preferring any more specific details of consensus NOS, to which the present study remains indifferent. Instead, this study adopts a certain ambivalence to specific epistemological stances and does not take stance on how Science of Science might support one or another version of NOS, or how it can be used to evaluate merits of different versions of NOS.

Science of Science (SoS) is a quantitative, data-driven research field that explores science through scientometrics (e.g., citation analysis), linguistic concept analyses, social network analysis and using big-data analysis, network methods and data-mining to analyse documents like research publications, reports and similar written or digitally-available documents (see, e.g., [35–41]). Such research shares the interest of history, philosophy and sociology of science (HPSS) in studying science, its structure and dynamics, and the factors that drive those dynamics. In that, SoS is complementary to HPSS in that it is quantitative, data-driven and focuses on large-scale phenomena of contemporary science, rather than being interpretative, history-oriented and making use of case studies. The SoS may significantly embody notions contained in NOS, and thus enrich NOS equally significantly as HPSS scholarship has done thus far. To clarify, the purpose of the present study is not to suggest augmenting NOS themes but to show that the existing scheme can be adapted perfectly well to accommodate themes arising from SoS; the findings and results emerging from SoS are material to enrich discussions through the lenses provided by NOS.

Here, I first briefly discuss the aspects of consensus NOS that are relevant to the present study and also outline in more detail the faultlines seen by its critics. Second, I outline Science of Science (SoS) as a research field. It is also informative to discuss why we may believe that the disciplinary structure of science and dynamics of science can be approached by paying attention to language, and how scientific communities use language and share terms and concepts within and between the communities. Third, in the most extensive section of this study, I summarise several findings of SoS that make contact with NOS themes. Finally, I briefly discuss implications of SoS for science teacher education and share some experiences from a recent university level course where SoS topics were discussed. With this study, I wish to convince educators interested of NOS to familiarise themselves with the findings and results of SoS, which might be a great source of examples relevant to NOS.

## 2. Consensus NOS: Some Supposed Fault Lines

In science education the most widely adopted viewpoint is consensus NOS that is based on views by Lederman, McClough, Abd-el-Khalick and their collaborators [4,6,8]. The consensus view on NOS is consolidated by seven tenets (see, e.g., refs. [5–7] and references therein): (1) the empirical nature of science, (2) the character of scientific theories and laws, (3) the creative and imaginative nature of scientific knowledge, (4) The theory-ladenness of scientific knowledge, (5) the social and cultural embeddedness of scientific knowledge, (6) the myth of (a single) scientific method, and, (7) the tentative nature of scientific knowledge. Here, theory-ladenness refers to how meaning and use of concepts are theory related and how that affects recognising and framing phenomena. The social and cultural embeddedness, on the other hand, refer to how science affects and is affected broader cultural factors and heritage, social fabric of society and its institutions as well as power structures and political and economic factors [1]. The initial NOS movement sought to dispel certain common views of science that it took as misguided and unfounded, for example: the immutability of scientific knowledge, and the existence of a single scientific method and its infallibility. To understand the nature of NOS and the debates around it, it is essential to understand that NOS is a science educators' view of science, distilled from views on science as scholars in the history, philosophy and sociology of science (HPSS) and science and technology studies (STS) have seen it. On that basis, consensus NOS is designed to support the formation of conceptions of science that serve the purposes of general education. Therefore, judgements about the acceptability of consensus NOS should not be grounded on how it compares to positions in philosophy of science (PoS) or to the views of practicing scientists, but instead how it manages to achieve its primary goals. Nevertheless, it is of interest to examine where the critics of consensus NOS have seen fault lines.

Already from at its beginning, the underpinnings of NOS were questioned [29] and debated [30,31,42]. Similar debates have emerged and faded repeatedly over the years (see, e.g., [9,13,21,25,26]). Critics of consensus NOS have often argued that it is too broad and based on overly vague notions, that it ignores important aspect of science and scientific knowledge (e.g., [11,13,28,43]), or even that it provides a distorted picture of scientific knowledge and the aims of science [9,15]. The critics of NOS often seek support from different versions of realism [9,13,15,21] and sometimes from more specific views borrowed from philosophy [19,22,23,25,26]. The critics have advocated views to augment or even to replace the consensus views of NOS by better founded or justified views, where support is sought from science philosophy, in some cases blending several varieties of realistic positions and semantic views on theory [9,13,21,25,26] or from very specific version of philosophical positions like critical realism [19] (see also [15,20,24]). As already pointed out in the early debates, such criticism ignores the purpose of general basic notions contained in NOS about which there is broad agreement within contemporary philosophy of science. The criticism pays too much attention to finer epistemological details of differing views and the professional discussions related to them (see, e.g., [31,42]). Moreover, the focus on such finer philosophical details has rarely produced practical teaching solutions. An exception is an alternative view to consensus NOS, known as the Family Resemblance Approach (FRA), suggested by Erduran and Dagher [22,23,27], based on the views of Irzik and Nola [25,26]. The FRA takes into account the disciplinary variation within sciences but recognizes that different scientific disciplines always have some sets of shared features; there is a family resemblance between and among disciplines. However, in closer look, focusing on how FRA becomes implemented in practical teaching, the outcome appears to be close to consensus NOS (see also [1]).

The criticism raised against consensus NOS and its imagined failures is nearly always raised on philosophical grounds, pointing out deficiencies when compared to adopted philosophical epistemological underpinnings, often some version of realism. Interestingly, the productivity of traditional epistemological issues (for example, truthlikeness and objectivity of knowledge, and questions related to realism versus constructivism) in understanding science has been challenged by the so-called philosophy of science practice (PSP), which turns away from metaphysically-oriented epistemological questions (see, e.g., [44,45]). While the reasons of PSP for its turning away from traditional epistemological and metaphysical positions may not be relevant for NOS, it reminds us that PoS does not offer unambiguous or self-justified positions to be taken as basis of NOS, but rather, different and varied lenses to view the sciences, each of them focusing differently but each also producing different distortions.

To escape from the ambiguity of philosophical positions, some critics have supported their arguments about the deficiencies of NOS with the results of case studies of practising scientists' views [10,11,16–19]. While this approach appears reasonable and productive at first, it has a recognisable weakness; scientists' views are so varied that nearly all positions discussed in philosophy of science can be found, but very little coherence (see, e.g., [19] for a review). Scientists' views provide a mixture of stances in which epistemic and ontological aspects are mixed, they are very context-dependent and weighted differently in regard to the practice of science (doing of science and scientific activities) and outcomes of science (body of scientific knowledge). Consequently, scientists' views reveal a mixture of realistic, constructivist, instrumental, pragmatic and anti-realistic views (see, e.g., [16,19]). This conclusion finds support from a recent extensive study of the views of practicing scientists regarding realism and its alternatives [46]. The study involved nearly 1800 scientist from seven fields of sciences, including physics, chemistry and biology as quantitative natural sciences, sociology and anthropology as qualitative sciences, and psychology and economy between these poles. In addition, five scholars from the field of history, philosophy and science (HPS) were interviewed and their views were compared with scientists' views. It was found, according to expectations, that there is indeed a clear difference in views between scientists and HPSS scholars; scientists favour realism more often than HPSS

scholars, who tend mostly to reject realistic positions. The situation, however, is not entirely that simple. Scientists also widely accepted the constructive empiricist position and the notion of empirical adequacy contained in it [46], and even clear anti-realistic stances if they were restricted to pragmatic aspects and practices of science instead of scientific knowledge as a product of science [46]. This again confirms the finding that scientists hold views that (from the point of view of more orthodox philosophies) contradict and even exclude each other. These notions provide enough basis to be cautious with regard to all arguments in which scientists' opinions are taken as guidance to form a picture of science; such a picture is certainly a canvas of scientist opinions but not necessarily useful to guide NOS at school level nor a as starting point for science teacher education. Rather than guidance to develop positions to understand science, scientists' views should be material to reflect on how positions based on PoS, HPSS and STS might help us to understand such positions and how they might or might not affect science and scientific knowledge.

Some authors have criticised NOS for assuming that science has an "essence" or "nature", and that such an essence can be consolidated in a form of scaffolding for the purposes of teaching [20,47]. A recent study that compared consensus NOS and one of its alternatives, Family Resemblance Approach (FRA) on NOS [22,23,27] raised similar critiques against them both [20]. It claimed that both consensus NOS and FRA are essentialist approaches on the nature of science in assuming that an essence characterising all science can be found, and dogmatic in the sense that they provide a single integrated framework to recognise that essence [20]. As a remedy, the authors suggested taking the viewpoint of Wittgensteinian language games to open up different windows on how the nature of science could be understood and discussed, and to give up the attempt to provide consolidated, integrated views in the form of lists or tenets. At a closer look, as visible in a response by Abd-El-Khalick [1] to similar critical remarks about dogmatism and normativity, such notions arise from assuming too much about (or reading too much into) the basic tenets of NOS. The very purpose of NOS is to provide practical approaches and designs for school-level teaching, where appropriate simplifications are needed. However, the level of sophistication can be raised when NOS is discussed at the level of science teacher education, when more in-depth discussions of the viewpoints offered by different positions become possible [1].

The critical evaluation of the underpinnings of NOS and FRA by do Nascimento Rocha and Gurgel [20] suggests that some of the faultlines seen by critics of consensus NOS are not necessarily very serious ones. First, NOS might benefit if it takes distance from the views offered by traditional philosophical accounts and their well-organised and neatly-arranged pictures of sciences. Instead, NOS should face and accept the complexity, unorganised nature, and plurality of science. Such a view opens up an approach that is more objective and less tied to a fixed and normative conception of how we should view science. Such a step also seems well aligned with consensus NOS, which emphasises the socio-scientific factors and social embeddedness of science, and how scientific knowledge is affected by these factors. As will be suggested later, a step in that direction could be taken using examples provided by Science of Science (SoS) but retaining the core ideas of consensus NOS to provide a scaffolding to discuss the findings of SoS. For a science teacher, SoS based on empirical evidence and concrete examples may help to embody the broad and general notions contained in consensus NOS. Second, NOS ought to turn away from traditional philosophy and its metaphysically-oriented discussion about the truth-value of knowledge or truth-likeness and endless debates on the preferability of some versions of realism over some version of constructivism. It is important to realise and understand that such different views exist and how science appears differently through such different lenses, because every one of them focuses differently. It is not, however, important or productive to select between the views or exclude some of them in favour of another; they can all serve to provide an understanding of science, and scientific knowledge and practices. Somewhat unexpectedly, giving up the many traditional questions related to the

epistemology of science and focusing on practices aligns better with the view provided by SoS than views based on traditional philosophical underpinnings.

### 3. Science of Science: An Acquaintance Deserving to Be Better Known

Many of the topics of interest for NOS, in particular the social and institutional factors of science, scientific inquiry and dissemination of scientific results, are explored by the research field called Science of Science [34]. The idea of Science of Science as a quantitative approach in exploring science itself goes back to seminal works by Garfield [48] and De Solla Price [34] and is today an intense data-driven field of study, drawing on the possibilities of data-mining and big-data analysis [35,37–41,49].

The Science of Science (SoS) is a quantitative study of sciences. The sources of data and information for SoS are scientific documents: academic publications, monographs and textbooks, as well as other written reports like research and patent applications. SoS mines connections between such documents based on citation analysis, textual analysis, and content analysis, for example of how vocabularies, terms and basic concepts are shared and adopted, and how they diffuse from one document to another. Structural connections based on such analyses are then taken as evidence of disciplinary connection, for example shared use of terms and concepts as cohesion of certain disciplines, unshared use of concepts as a sign of different disciplines [37–41,49]. Similarly, in citation analysis, recurrent collaboration patterns are taken as signs of collaboration and shared research interests, publications in shared forums as signs of institutional cohesion of the community. SoS touches on many important social and institutional factors that affect the formation of scientific communities [37,39,50].

Over the last two decades, the developments of big-data analysis and data-mining have opened up unforeseen opportunities to expand the scale of SoS and to deal with millions of documents. Methods to explore the vast data sources are rapidly increasing and becoming ever more sophisticated, producing a valuable source of evidence-based results about the structure of science. In exploring the documents, SoS makes use of data-mining algorithms, network based methods of analysing the relevant connections, and relationships in the data [37,39,50]. One important output of SoS has been different maps of the disciplinary structure of sciences, which provide a panoramic view on the universe of different sciences. Such maps are invaluable for providing an overall picture of the landscape of science and how different fields are related, and they show very clearly the nested disciplinary structure. The maps also reveal internal dynamics of the sciences by showing how disciplinary boundaries have changed, appeared and disappeared, and how new disciplines have emerged (see, e.g., [36–39]).

As a quantitative, data-driven research field, SoS is largely independent of the philosophical underpinnings one often finds in HPSS, where interpretative viewpoints are often motivated through considerations based on philosophy of science. Nevertheless, it is of interest to note that the approach of SoS, in paying attention to linguistic and semantic structures to understand science, its structure, and the communities within it, aligns well with Thomas Kuhn's conception of science, which he developed after their well-known work "The Structure of Scientific Revolutions" [51]. In their later research [52], the focal point of their views shifted to linguistic structure [53,54]. On the basis of importance of linguistic structures, Kuhn introduced lexicons (of scientific language), which became central to their views of scientific communities and thus, of the disciplinary structure of science that they constituted. The systematic and correct use of lexicons characterises scientific communities, not its individual subjects. The correct, systematic and normative use of lexicons, on the other hand, is revealed by how the community approaches its basic problems and uses the lexicons in problem solving, and how it instructs its members and newcomers using lexicons. [52–54]. Kuhn's view of the importance of language and lexicons in shaping science, its communities, and even its worldviews, owes much to Wittgenstein's language games (see [53,54]). Such views, aligned with SoS, would be an

interesting direction in which to further push the ideas of how NOS could make better use of Wittgensteinian language games, as suggested by some researchers [20,47].

SoS, however, does not derive its motivation and underpinnings directly from any specific philosophical views like Kuhn's or Wittgenstein's (which are only seldom mentioned in the context of SoS) although we may recognize paralleling views in how SoS too pays attention on how science is communicated and how the use of language of science provides identity and cohesion for scientific communities, and how language and its use is present in discussing, framing and solving problems and deciding what is relevant and worth effort, resulting eventually in the production of scientific knowledge. Such focal points of course will miss many of the interesting and insightful discussions HPSS can provide by using underpinnings grounded in philosophy of science, sociology of science and science and technology studies. Nevertheless, despite the limitations that come with such an approach, SoS provides a treasure trove to enrich and enflesh NOS. It may provide significant support for NOS in opening up views of science that are complementary to HPSS and focus more strongly than HPSS on a panoramic picture of contemporary science. Therefore, SoS is a field that deserves to better known and appreciated as part of NOS.

## 4. Science of Science Themes for Embodying Nature of Science

Contemporary Science of Science (SoS) provides many access points to NOS-related discussion and the kind of raw material and evidence that needs to be embedded within NOS themes and viewpoints. On the other hand, in this way SoS helps to embody the general and idealised notions contained in NOS. For example, discussing the need for disciplinary-specific approaches to NOS (at the level of higher education and teacher education) becomes more fruitful when one has basic information on the features of that structure, its dynamics, and the factors affecting the dynamics. In addition, questions about the theory-ladenness (meaning theory relatedness of concepts and how phenomena are framed) and role of theories and experiments can be made more concrete through examples of how the relationship between theoretical and empirical research appears in the light of SoS. Many similar valuable ways are now available from the vast and rapidly-expanding literature of SoS. In what follows, some themes and possible sources are discussed. Later, a short summary is provided on how the themes discussed might fit in with NOS themes.

### 4.1. Disciplinary Structure of Sciences

The disciplinary structure of scientific knowledge is an extensively researched topic in Science of Science, and the results with regard to the disciplinary structure show not only the clear structure of disciplines, but also a certain stratification within it, as well as cultural and societal country-related differences in disciplinary profiles. Therefore, the key question is not about the existence of that structure, but about its form, how one can get information about it, and whether different sources of information provide a different picture. In Science of Science literature, the disciplinary structure is discussed in many studies and from many different viewpoints, for example: techniques for mapping structure [41,55,56]; disciplinary layout [57–59]; structural similarity [60,61]; and interdisciplinarity [62,63].

The macrolevel analysis of the structure of sciences reveals a clear disciplinary structure, which is a starting point for understanding the nature of science and the dynamics that shape the disciplinary structure. The analyses of disciplinary structure are most often based on citation and scientometric analyses, and include sciences from medicine to mathematics [36,39,40]. Similar studies with very similar results are also obtained by focusing on shared concepts and terms [64]. In such analyses, different areas of science, from biological sciences and medicine to physics and mathematics, fall into clusters with clear borders; that is, disciplinary clusters. The details of the structures and substructures may differ depending on the method of analysis and its sources, but all studies lead to rather similar outcomes regarding the disciplinary areas: medicine, biomedicine, biology in one cluster; physics, astronomy and earth sciences in another; electrical engineering and engineering in their cluster and so on, with tighter connections between closely-related areas.

In a study by De Domenico et al. [36], covering one hundred years of sciences, researchers' activity in publishing in certain disciplinary fields is investigated in detail. The results of the study show how different disciplines were rather closed a century ago, not interacting so much with other disciplines and with researchers contributing mainly within their own areas. Border crossings started to become more prominent only 50 years ago, first with a clear flow of knowledge between the areas of medicine, biochemistry, genetics and molecular biology, and on the other hand, between disciplines in a cluster of physics and astronomy and earth and planetary science, as well as between the disciplinary clusters of chemistry and chemical engineering. Interestingly, however, some areas like nursing and health professions already had open boundaries a century ago [36]. Today, the flow of knowledge across borders is common between practically all areas. Most disciplinary areas have had an initial stage of development where they stay rather isolated, but eventually they develop more open boundaries with other disciplines, with strong flows of knowledge across the boundaries [36]. The disciplines of medicine, physics, astronomy, chemistry and mathematics, however, have stayed more isolated than some other fields. On the other hand, some disciplines like computer science and environmental science have developed over the last decades from isolated disciplines to ones with remarkably open boundaries and significant flows of knowledge to other disciplines. This very central role of computer sciences is obviously related to recent public investments in multidisciplinary research on artificial intelligence, which has boosted the flows of knowledge between computer science, mathematics, cognitive science, philosophy of mind and electrical engineering [36]. Such examples show that while the disciplinary structure can be resolved quite clearly, there is increased flow of knowledge between the boundaries without the tendency for boundaries to dissolve or become invisible.

The case of medicine and physics is interesting on closer inspection. Although these disciplines appear as isolated when the mobility of authors between the disciplinary areas is inspected, their contribution to overall flow of knowledge is remarkable when flow of knowledge and use of knowledge is in focus. This viewpoint shows that, while researchers in these areas stay within them, the results are widely used in different contexts, but knowledge also flows from other areas of those disciplines; medicine and physics act as sources and sinks of knowledge in the bigger picture [36]. This is apparently related to the wide applicability of techniques and methodologies developed within these fields. A similar notion comes up on the basis of investigating the co-occurrence of shared concepts in books and monographs. Such results too show the importance of medicine and physics as well as chemistry and mathematics to all branches of natural sciences, demonstrating the importance of those sciences for many other disciplines [65].

### 4.2. Subdisciplinary Structure: Physics and Chemistry

A gradated subdisciplinary structure is found also within disciplines. Within disciplines, the isolation and importance of subdisciplines may also vary significantly. Such a situation is exemplified by detailed studies of disciplinary substructures in physics [66–69]. Physics as a discipline is an interesting case because of its role, as seen in the previous section, in connecting and underlining many other disciplines. As noted for example by Sinatra et al. [67], one reason for this role is that physics has always been in dialogue with many other disciplines, especially mathematics and chemistry, and that the dialogue has been driven by methodologies that transgress the boundaries.

An extensive study by Sinatra et al. [67], based on about 5 million published papers in physics between 1900 and 2012, show that it is possible to distinguish a core of physics papers (2.4 million) and interdisciplinary non-physics papers (3.2 million) that refer to that core and are of interest for physics. Interestingly, within the non-physics papers, six physics Nobel-prize winners can be identified. This demonstrates strikingly the important role of physics in other sciences. A closer examination of the papers in the physics core shows the important role of quantum physics in emergence of many new fields, from condensed matter physics to nuclear physics. Since then, the growth of papers published in physics

has been nearly exponential. The growth was especially rapid after the Second World War, and settled down in the 70s to the current rate, at which the number doubles in about 19 years [67]. However, as pointed out in another study [70], the exponential growth of literature does not mean that new ideas and conceptualizations increase similarly; rather, the increase in cognitive content (as measured by terms indicating new concepts and conceptualizations) has since 1930s increased only linearly and actually slowed down after the 70s. As Sinatra et al. points out, the growth rate of publications in physics is an outcome of societal needs, access to resources, and simply an increase in the number of researchers.

Physics was up to the late 1930s mostly an isolated field of study and also quite shortsighted in noting literature even in its own field, with references extending only some years to previously published results [67]. A change started to take over in the late 1930s, with a rapid increase in publishing physics-related results outside the physics core literature, indicating the importance of other fields to physics and, in reverse, the importance of physics to other fields. This change was partly, but not entirely, related to the rising importance of the new quantum physics. However, in the 1960s a period of stronger isolation of physics and in subdisciplines within physics set in, but soon after that, gave way to deeper and broader attention to fields outside but also within physics. The explanation suggested by Sinatra et al. for this change is a change in reviewing practices, which changed in broader scale from editorial acceptance to peer-review, and subsequently through peer-review, forced researchers to note results in related fields more broadly. Interestingly, according to Milojevic [70], in physics the late 1960s and subsequent 1970s was a period of lower publication rate growth, but a rise in the number of new ideas and concepts. Such dynamics and changes in dynamics where changes might be triggered by academic practices and changes in norms, and where growth in the number of scientific results does not correlate with an increase in conceptual extent, are important examples of the complex dynamics of sciences. It also warns against adopting simple pictures of a cumulative, ever-increasing body of knowledge, which grows only by its internal logic and dynamics.

Bibliometric analyses reveal that the disciplinary substructure of physics consists of three large communities or clusters: the first cluster consists of condensed matter physics and interdisciplinary physics and related fields; the second cluster involves electromagnetism, atomic and plasma physics; the third cluster contains particle, nuclear and astrophysics [67]. Within a given cluster, the fields cite subfields within them frequently, but not so often subfields that belong to other clusters. Between the subdisciplines, it is possible to even find citation barriers, showing a strong tendency to a certain isolation of the subdisciplines. Among the subdisciplines, particle and nuclear physics appear to be the most isolated subfields. General physics, however, has a special role since it unites all these three fields. The emergence of such disciplinary clusters is much as expected, but it is interesting that citation analysis reveals the existence of clusters so clearly, as demonstrated in the study by Sinatra et al. [67]. Similar results are also obtained in other similarly focused studies, where roughly the same subfields are recognised [66,69].

The connectedness and isolation of different physics subfields are interestingly related to the lifetime of scientific publications in different fields [67]. The lifetime of publications is at least partially related to the timescales of change and evolution of scientific knowledge within the disciplines. Old knowledge is not necessarily abandoned or entirely changed, but incorporated as part of new knowledge, transformed, and improved. In physics, the general physics, classical physics (in particular electromagnetism and optics), and interdisciplinary physics have great impact on physics in general. These research areas also have the longest lifetime of scientific outputs, and thus the longest duration of impact (from 11 to 14 years on the average, some publications exceeding the average significantly). On the other end of the spectrum is nuclear physics and particle physics, where new research becomes forgotten or fades away after 6–7 years [67]. Clearly, publications coming from insular areas that are not central for other areas of physics have not only smaller impact but also shorter lifespans.

The existence of the clear boundaries between the subfields in physics is a feature of scientific knowledge, not a feature of doing science. The picture of disciplinary boundaries changes when one focuses on scientists and on how they migrate from area to another. According to Battiston et al. [66], most physicist start their careers in the three subfields but do not stay in these areas for all their careers [66,67]. The specialisation in single fields is not a rule of a typical career in physics and the majority of physicists (63%) work in two or more subfields over their career. As found by Battiston et al., only 1% of researchers in interdisciplinary physics are specialised in it, in condensed matter only 42% and in high energy physics and nuclear physics, 34% and 25% respectively. Condensed matter physics is a starting point for many physicists who later work in interdisciplinary or general physics. One reason for this may be the role of statistical physics, which is part of the condensed matter physics cluster and is widely applicable in many other fields. The clusters of high energy and nuclear physics, on the other hand, feed physicists to astrophysics, probably for reasons related to the expertise in radiation physics needed in all these areas. Given that high-energy physics and nuclear physics are among the most closed disciplines, this is an interesting finding. The explanation provided by Battiston et al. is that migration between fields is related to transfer of know-how and methodologies, especially in the case of condensed matter physics, and to skills of working in large teams and on long-term projects, as is typical in high energy physics. Physicists working different subfields apparently combine expertise of those fields, thus facilitating invention of new ideas, combination of ideas and approaches, all features needed for discovery of new ideas [71,72]. In addition, regarding the central roles of condensed matter physics and high energy and nuclear physics as the starting points of careers, Battiston et al. [66] note the historical developments of the research strategy of western countries, which focused on these areas strongly during the 80s. The investments made several decades ago are still structuring institutions and educational systems, although the needs of research have changed since then.

Chemistry provides another interesting example of evolution of subdisciplines, their emergence, waxing and waning or submerging of some special areas into new growing, more dominant areas. A recent study by Waaijer and Palmblad [73] explored how chemistry as a field of research has evolved from 1929 up to 2013 as it appears in one journal, Analytical Chemistry. Although only single journal was used to monitor evolution in chemistry, the relatively broad scope of the journal is assumed to provide useful insight on the development of the field. The study by Waaijer and Palmblad used content sensitive methods, based on detection of key words and they co-occurrence to recognize subdisciplinary areas and their evolution. In the era of focus, in chemistry, as in other natural sciences, the research output increased exponentially. According to Waaijer and Palmblad [73], the most important areas within chemistry in 1929–1940 were related to instruments and apparatus, gases, inorganic chemistry and applications of chemistry. The analytic chemistry, on the other hand, was one of the most important and central areas from 1929 to 1990 but in period 1991–2000 it begun to merge increasingly with electrochemistry and sensor technology. In the early years of 20th century chemistry, gravimetric and volumetric experimental methods have had the most important role in experimentation, gradually shifting to background with advent of new methods, but never completely disappearing; in chemistry, similarly as in physics, old methods do not necessarily die out but are incorporated as routine parts of novel methods, thus becoming invisible.

Topics that have developed over time include electrochemistry, chromatography, and mass spectrometry. From 1941, electrochemistry begins to appear in publications, connected most often to inorganic chemistry and metals, but during 1951–1960, it appears as self-standing subdiscipline. Similar development is found in spectrometric methods, where chromatography begins to stand out from period 1951–1960 onwards, and mass spectrometry after period 1971–1980. Mass spectrometry develops through its connection to chromatography up to 1990, and after that, it forms its own subdiscipline. An example of completely new discipline appearing in 2001 and growing after that is microfluidics,

which is strongly connected to theory and simulations. Consequently, according to Waaijer and Palmblad [73], chemistry as a research field has changed considerably from 1929 to 2012, the major change being the decrease in chemistry based analytical methods and increase of physics based analytical methods, especially the increasing importance of mass spectroscopic methods.

In summary, the analyses of disciplinary structure of science provide ample evidence that to understand nature of science, one needs to pay attention to how disciplinary structures emerge, how borders are crossed and how border crossings shape sciences, how some structures are dissolved and others become insular. Such changes are inherently connected to historical traces of how disciplinary identities are formed and on the other hand, social and political guidance and the rise of new needs to which science needs to attend. One important factor affecting the formation and change of disciplines is the migration of scientists with different know-how on problem solving and use of methods. In short, such factors provide important and concrete evidence of the strong societal factors in doing science and how societal decisions affect the structure of science. Science of Science does not attempt to provide an overall, simplified and streamlined picture of sciences, not even in cases of its disciplinary substructures. Rather, SoS invites us to accept the complexity as a genuine feature of science

*4.3. Theory and Empiry: Physics, Chemistry and Biology*

The relationship between experimental and theoretical physics provides an interesting viewpoint on the roles of methodological boundaries. The discussion about the experimental-theoretical division in physics is a long-running one, but on quite an informal and subjective basis, and only recently have evidence-based views become possible through scientometric analysis [68]. A study that explored the connections of experimental and theoretical research in physics found that these two fields are very closely connected and the borders between them are easily crossed; there is no deep division between experimental and theoretical physics [68]. The analysis showed that experimental physics and theoretical physics are very tightly connected, with substantive flows of knowledge transgressing the boundaries (but with boundaries existing). In addition, the citation patterns, showing how knowledge is used and distributed, are remarkably similar within and between the areas of experimental and theoretical physics [68]. Interestingly, such homogeneity can perhaps be attributed to a certain similarity in ways of doing science, reporting it and augmenting for knowledge claims; all these features are closely connected to norms and conventions within a discipline.

This picture of relations between theoretical and experimental physics, however, changes when top-cited authors are the focus of attention [74]. In this case, it is clear that the top-cited theorists are much more visible in the literature than experimentalists are. The reason for such a situation is that many key theoretical results are of interest and importance in many other subfields of physics, as is expected in science, which is stratified in the sense that one can recognise a fundamental theoretical knowledge that underpins most of its areas [74]. Curiously, this might also be related to the high visibility of theoreticians in popularising physics and its recent advancements.

A close connection between theory and empiry is found also in chemistry, in particular in its subdisciplinary areas of physical chemistry and analytical chemistry. In physical chemistry, the atomic physics and quantum theory have close connection to experimental research of molecular spectroscopy and reaction kinematics, theory being involved not only in guiding experiments but also in devising new experimental techniques [73,75]. According to Johnson [75], in physical chemistry experimental and theoretical methods are today converging, in particular in modern theoretical methods of quantum many-body systems and computational to investigate chemical bonding [75]. Another research area where on finds rapid convergence of theoretical and empirical methods is found in nanoscience, where the relation between physics and chemistry is close and fluid [76].

Close connection between theory and empiry is perhaps quite expected in physics and chemistry, where mathematisation and use of advanced experimental methods are central and owing to historical development of the disciplines. However, similar convergence of theory and empiry characterise also "softer" science, biology. In biology, in much the same ways as in chemistry and physics, the theoretical models are often mathematical models, which are used to formalise hypotheses, simplify the complexity of phenomena, or integrate phenomena on different temporal and spatial scales. It is also expected in biology that theoretical models can be related to empirically observable phenomena, if not directly, at least at a certain level of abstraction and idealisation [77,78]. A recent survey [78] reported that many researchers in ecology and evolutionary biology think that theoretical and empirical research should be integrated in biology for the advancement and success of the field. However, only a fraction of the biologists share the view that such integration has taken place or been successful, while the vast majority sees theoretical and empirical research as separate and having very little interaction [78]. This view of scientists (view of practicing scientists) was challenged by a recent scientometric study, which shows that in fact the integration and interaction of theoretical and empirical research in biology is more common than might be expected on the basis of the survey. It is evident that in biology (or in the branches of biology explored by the study), the majority of the research is done within subfields (as in physics), but a significant fraction (about 20%) transgresses the boundaries of subdisciplines; there is significant knowledge flow. What is important, is that 60% of research outputs were theory or empiry insular (self-citing only papers that were theoretical or empirical), while 40% integrated theoretical and empirical research. This is more than could be expected from a survey based on the views of practicing biologists. The authors of the study raised the question of why there was such a discrepancy between the views of practicing scientists and the results of scientometric analysis. As one possibility, they recognised the way the theoretical work is used as background and to provide motivational underpinnings for the empirical work, rather than explicitly integrated as part of the empirical work. Another reason pointed out in the study is the role of review-type contributions that explicitly attempt to create bridges between theoretical and empirical work.

When the roles of theory and empiry are compared in physics, chemistry and biology, it is clear that there are certain differences in why theory and empiry are integrated and how the relation of theory to empirical research is conceived. On the other hand, there are many similarities, which arise partly from how the empirical work becomes motivated and guided by theoretical assumptions even in the cases that it does not explicitly integrate theory. In both cases, however, the relationships revealed by scientometric analyses and SoS embody the science philosophers' notion about the theory-ladenness of science. On the other hand, the findings warn against making oversimplified assumption about the primary role of empirical research as the driving force in the advancement of science.

### 4.4. Cognitive Extent and Knowledge Growth

The volume of scientific knowledge, as measured by the number of publications, reports and other written scientific documents, has increased exponentially over the last century, with a yearly growth rate of 5% (see, e.g., [79]). This, however, does not necessarily mean that new ideas and significant scientific findings, i.e., the cognitive content of science, have increased equally. Here, cognitive content (and content) refers to new concepts, terms, theories, models and such intellectual or artefactual (i.e., devices) that can be recognized by using data gathering methods of SoS [70,80], as outlined in Section 3. Similarly, it is often mentioned that today large collaborative groups are the most significant in producing scientific output (see, e.g., [80–82]). It is claimed that big groups are becoming more common in all areas of science, and also that they are responsible for the main intellectual output of scientific knowledge (see, e.g., [72,81,83]). The high visibility of large groups in producing many scientific reports, however, does not necessarily mean that large groups

also produce the majority of the new ideas and insights, i.e., the new cognitive content of science [70].

The growth in the cognitive content of science and the role of groups of different sizes in production of the new cognitive content has recently been questioned by taking a closer look at how new concepts and ideas are introduced in scientific publication, by using not only citation-based analysis but also deeper textual analysis of content [70]. In an extensive study, Milojevic [70] explored the cognitive content of articles in physics (440,000 articles), astronomy (160,000 articles), and biomedicine (19,600,000 articles), which were published over periods of 117, 125, and 67 years, respectively. The results of the study show that although the growth in productivity in all these research areas was indeed exponential, the cognitive content of the fields increased only linearly rather than exponentially. In all these areas, the growth of cognitive content is within a factor of a few, while the publication volumes rose a hundred- to a thousand-fold [70]. An interesting example of the disconnect between productivity and cognitive progress is provided by physics, where in 1970s the growth in production was low, nearly stagnant, but despite that, it was the period of fastest cognitive growth [70].

The role of groups of different sizes appears also very differently in an analysis based on content, instead of on mere citations. According to Milojevic, in physics and astronomy, the publications by single authors or pairs of authors produce the majority of the cognitive content, and in the case of several authors, more authors indicate less cognitive content. In fact, publications produced by very large teams (i.e., groups working in big science) provide only 35% of the cognitive content of physics. This notion, even when the limitations of the study are taken into account, puts the claims of the dominance of big science groups in a very different perspective. The differences of group size with regard to the cognitive content and its extension can be at least partially understood on the basis of how groups are formed. Formation of small groups, involved with the greatest breadth of cognitive content, appear to follow a random process (Poisson process), while larger groups grow by agglomeration and take advantage of cumulative growth. Large groups are also more specialised than small groups, and thus do not cover the entire research field within their disciplines. It is also evident that small groups focus mostly on theory, and thus are involved in the production of novel conceptualisation. A plausible interpretation is that small groups are involved in producing new concepts, not all of them of lasting value, while big groups are involved in testing and justifying the knowledge, but not taking so many risks in proposing tentative concepts, models and theories.

This interpretation of the different cognitive roles of small and big groups finds support from a study by Wu et al. [84], which analysed 65 million papers, patents and software products from the period 1954–2014 to identify how group size and the introduction of new ideas are correlated. The results show that small groups indeed tend to cause disruptions, by adopting and introducing ideas that are out of the mainstream, whereas big groups develop the existing ideas further. A consequence of focusing on existing mainstream ideas is that the work comes to the attention of others more rapidly, which also explains why research done in large groups and teams gains better visibility. The research done in smaller groups gains attention much later or not at all. The importance of this finding about the different roles of small and big groups is that it guides attention to the need for intellectual diversity and diversity in working modes. The advancement of science needs both kinds of activities, feeding new ideas, not all of them fit for further life, and testing and developing the existing ideas further. In discussing science, it is important to keep in sight such intellectual diversity instead of emphasising the intellectual dominance of big groups and big science, which may result if only volume of productivity and activities are valued.

### 4.5. Interdisciplinarity and Knowledge Flows

The advantages and disadvantages of the disciplinary structure for advancement of sciences are currently much discussed, with conclusions that favour interdisciplinarity

for the advancement of sciences [62,63,71,82,85,86]. However, the degree to which inter-disciplinarity is realized, and even its benefits, are questioned and disputed [87]. It is a common claim that contemporary science is strongly interdisciplinary and disciplinary boundaries are dissolving. However, this claim, if taken at face value as indicating that boundaries are dissolving, fails to find strong support from quantitative research of science. Rather, the overall picture is more complicated, with clear disciplinary structures visible but significant border-crossing between boundaries that remain instead of dissolving [85]. It is thus necessary to acknowledge the diversity and existence of the stratified disciplinary structure in order to understand the role of disciplinary structure, how it emerges from different research strategies and methods, and how it affects the development of research agendas, researchers' careers, and the flow of knowledge.

Disciplinary boundaries and the stratified structure of disciplines, however, do not hinder knowledge flow and diffusion. On the contrary, extensive evidence flowing from Science of Science studies show that strong connections exist between the boundaries, and researchers constantly export and import ideas from one disciplinary area to another [71,82,85,86]. The knowledge flows occur not only between different disciplines; there are significant and important knowledge flows between sciences and areas where their knowledge is consumed. Consequently, researchers of Science of Science have explored such dynamics of knowledge production and consumption from the perspective of the exchange balance of scientific knowledge [88] and also as "food webs" of knowledge; as dynamic networked systems [89].

One of the most studied areas is knowledge flow from science to technology, as it appears through the patent-to-paper citation data. Such connections are interesting because they arise from collaboration between pure sciences and technological industry, on the one hand related to knowledge transfer in practical problem solving and on the other, also closely related to innovation and the commercial success of nations [90–93]. It has been claimed that rapid advancements in information technologies facilitate knowledge diffusion and transfer, and thus effectively diminish the effect of geographic barriers [94]. However, the recent results based on scientometric analysis do not necessarily support so a clear-cut interpretation.

The paths leading to scientific innovations involves bidirectional diffusion of knowledge and knowledge transfer between science and technology. Together, science and technology decide the direction of scientific advancement and progress, and they are inseparable parts of knowledge evolution. The research in Science of Science has provided much information on the paths along which this knowledge diffusion and transfer occurs [95,96]. In a study focusing on knowledge flow paths between science and technology, it was shown that there are significant similarities but also differences between the evolution mechanisms in science and technology.

### 4.6. Disciplinary Structure at Country Level

The disciplinary structures of sciences exhibit interesting variations from one country to another. For example, the scientific activity of the West European countries has a different disciplinary profile compared to the post-communist countries of Central and Eastern Europe, as well as compared to Asian countries [64,97,98]. Roughly, the Western European pattern of disciplinary structure can be characterized as being strong in clinical medicine and biomedical research, while the post-socialist countries and China are strong in chemistry and physics. However, in the Western European profile it is also possible to find clear biases: for example, the UK is stronger in medical and life science than Germany, while Germany is strong in physical sciences and engineering [64,97]. It has been pointed out that it is possible to recognize historical and ideological reasons behind such developments in disciplinary specialization [64,99]. It is interesting to note that the disciplinary structures of different countries are indeed different, that there is a connection with the cultural and historical heritage of the countries, and change is slow.

Differences in disciplinary profiles are also reported in studies that compare G7 countries to BRICS countries (Brazil, Russia, India, China and South Africa). In the G7 countries, life sciences play the most important role, while in the BRICS countries physics, chemistry, mathematics and engineering are the strong areas of research [61]. The disciplinary structure of European countries is consolidating to a profile with certain identifiable European characteristics, with many developing countries approaching that same profile but some others clearly departing from it [60]. It is obvious that national strategies on investments, including decisions to funnel funds to certain areas in R&D to improve technological competitiveness and the economy, affect the disciplinary profiles, as does the simple fact of geographical location, through its effect on trade and commercial collaboration and competition. Such factors directly affect the disciplinary structure, because the disciplinary structure has a strong effect on competitive positions in economy and technology [58,100].

Some other studies have focused on the interaction and balance of knowledge flows between different countries [88,101], paralleling the studies that have explored the disciplinary structures and profiles in different countries. These studies have pointed out the interesting phenomena of obvious localization; despite the global nature of science and pervasive use of IT technologies, science is still in many respects local. The results of such studies show that geographical distance is quite significant on the national level, and affects the knowledge flow on the continental level as well, but becomes insignificant on the international, intercontinental level. The role of geographic closeness in knowledge flow can be attributed to the importance of direct personal interactions and social effects, like connections with mentors, the effect of schools in the research field, and on the domestic level of biases in citation practices favouring domestic research [88,101]. Other studies have reported similar results, showing that citation links decrease with increasing distance [83,86].

In summary, while the studies of the dynamics of knowledge flows end up with different, if not contradictory results, they open up discussions of the complexity of the knowledge flows and point out how different factors eventually contribute to the formation of the successful paths. This is a clear example of the mutual dependence of science and technology, and how they are both related to the economy and economic competence of nations. Such information is a good starting point for discussions of the societal effect of science and how society affects science.

*4.7. Other Topics Covered in SoS That Are of Interest for NOS*

Science of Science provides a vast and rich stockpile of information about the disciplinary structure of sciences, detailed maps of subdisciplinary structure, the cognitive content of sciences, community and collaboration structure, interdisciplinarity and transgression of disciplinary boundaries, country-related differences, and knowledge flows. Some of these aspects have been discussed here, but many interesting and important topics of interest that are needed to understand science are yet untouched. Two additional topics that are clearly of interest for NOS and on which SoS provides much information are related to discoveries and institutional practices.

Discoveries and originality of ideas have been the focus of some recent studies [102,103]. One way to detect originality and new ideas rests on textual and citation analysis, on how concepts or ideas are available in previous literature [102]. However, the detection of landmark papers as based on expert opinion does not necessarily support the view that such citation-based analysis is reliable in detecting originality. Consequently, several novel methods to detect originality and novelty have recently been explored.

The institutional practices of distribution of scientific knowledge and securing the reliability of knowledge through different quality control means is another important aspect of science. Peer-review of scientific publications is perhaps the most important single process in securing the quality of scientific outputs. Peer-review, however, is far from unproblematic, and how its internal dynamics may bias the type of results published has been extensively discussed [104]. It may have astonishingly low reliability with regard

to the importance of the results [105]. Interestingly, the peer-review system also affects the formation of research collaboration and networks [106].

The above are only a few examples discussed extensively in the SoS literature, and all of them are quite evidently also of great interest for NOS. Other topics of interest covered in SoS literature are: science and economic growth, funding strategies and policies, scientific careers and career opportunities, and science and entrepreneurship. In addition, topics related to the development of scientific fame and success are of interest for NOS, at least at the level of higher education, in the education of science teachers, when the broad notions contained in NOS are in need of concrete examples and deeper discussion. Therefore, it is highly recommended to use the information provided by SoS to embody the very general and non-specific notions on which NOS is often based.

## 5. Topics of SoS as Material for NOS Themes

The topics addressed by SoS and the information it provides can be quite well discussed in science education as self-standing topics, from SoS perspective and without connecting them to NOS discussions. However, there are many rather obvious connection points to NOS themes and making such connections may be advantageous. The themes provided by SoS are not meant to replace NOS themes, nor to add new focal points or viewpoints, but to provide information and evidence that can be used to embody the general and idealized notions of NOS. In that, the role of SoS parallels with ways to acknowledge the information based on HPSS scholarship, traditionally having had a central role in NOS, and recent studies about practicing scientists' views [10,11,16–19], which are also acknowledged to be important contribution to NOS [1].

The scholarship in HPSS has already provided and continues to provide embodiments based on historical examples, as well as case studies related contemporary science. Such analyses are invaluable but limited. The examples based on history of science, looking back decades or even centuries, help us to understand how we have arrived at the present stage of sciences, but provide limited insight on current developments and their dynamics. Moreover, HPSS case studies based on contemporary science are often too detailed and too context-dependent (specific examples within a subdiscipline of a certain disciplinary area of science) to provide panoramic pictures at the level of the generalizations needed for NOS.

The practicing scientists' views [10,11,16–19] provide also insights on doing science and how scientists themselves see science. However, the practicing scientists' views cannot be priorized in guiding how NOS sets its goals, because the views lack coherence and are obviously most often personal reflections rather than outcomes of scholarly studies (see, e.g., [1,32,33] as well as [46]). In using information flowing from SoS one encounters a situation paralleling the case of practicing scientists' views, when such information is embedded as part of NOS. The information flowing from SoS should not dominate in setting goals for NOS, but neither, it should not be ignored but embedded. It is important to acknowledge that neither practicing scientists' view nor SoS can have discriminating role in making preferences between different varieties of NOS; it can adopt different underpinnings flowing from different epistemological backgrounds and evaluations of what is central in science, all of them viable. A decision to adopt a certain version of NOS is a question of applicability in level of use of such views, didactical practicality and choice of focus of teaching, not a competition of correct epistemological background thinking about science.

In what follows, I discus how SoS can be utilized as a source of panoramic views and vast amounts of facts about many areas of sciences. The appropriate NOS embeddings of SoS are here discussed through NOS themes as summarized in consensus NOS [1]. This choice is not meant to priorize consensus NOS as a preferred version of NOS. The solution is practical in sense that in the level the uses of SoS to NOS are discussed here, it is unnecessary to underline the differences between consensus NOS and other variants of approaches (like FRA). In parallel with notion that different versions of NOS do not

differ too much in level of education relevant for schools [1,6], it is also concluded here that differences between NOS variants can be ignored on discussing relation of SoS to broad thematic categories of interest in NOS. Consequently, although the embedding of SoS topics to NOS themes is now provided by utilizing categorizations from consensus NOS, similar embedding should be possible for example for FRA or some other version of NOS. Many of the topics that SoS can provide for NOS fall into categories related to disciplinary structure and its dynamics, theory-empiry relationship (including model-based views) and social and institutional factors (e.g., as discussed in FRA, see Erduran and Dagher [22]). As a result, embedding the themes available from SoS to FRA appears rather to restructure NOS themes under the FRA categorizations rather than having significant consequences on how the SoS themes can be utilized if FRA is chosen as preferred approach on NOS. In addition, some factors as they come out from SoS cannot be factorized in such a categorical way as consensus NOS suggests, but this is probably not a problem, because, its categorizations are not meant to be fixed but, at least at a higher level, to form a connected set of themes [1]. Similarly, factors related to the role of models, theories and laws in SoS are not as explicit as introduced in FRA (see Erduran and Dagher [22]). Such knowledge structures as they appear in SoS in different disciplinary areas, can quite well be discussed from model-law-theory viewpoint offered by FRA, but without necessity to pose such structures as an overarching organizing system in disciplinary areas, where such divisions may be difficult to recognize or appear to impose too forced pre-fixed structures (see discussions, e.g., in [44,45]). This flexibility of choices to embed topics from SoS to NOS is result from that SoS is equally indifferent to philosophical and epistemological stances on science as practicing scientist themselves are (see Beebe and Dellsen [46]). This, of course, does not entail that for education NOS should be reduced to descriptions flowing from SoS. On the contrary, different variants of NOS provide valuable lenses to examine and discuss such descriptions.

In what follows, the themes discussed in Section 4 are now aligned with thematic categories of consensus NOS as outlined Abd-el-Khalick [1], from point of view of teacher education. This alignment is summarised in Table 1. In addition, a reference to pre-service students' notions concerning NOS themes, as they are discussed in next Section 6, is provided for better clarity of paralleling the themes. The overlap of categorization in Section 4 (reflecting the themes as discussed in SoS), in Section 5 (reflecting ordering in NOS) and in Section 6 (based on students' notions) is not perfectly aligned, but nevertheless, has much overlap.

**Table 1.** Alignment of themes as discussed in different Sections 4–6. For Section 6, numbering refers to grouping of pre-service teachers' feedback.

| 4. SoS | 5. NOS | 6. |
|---|---|---|
| 4.1. Disciplinary structure | 5.1. Discipl. strct. & knwl. flow (4.1, 4.5) | 1,2 |
| 4.2. Sub-disciplinary structure | 5.2. Durability and change (4.1, 4.2, 4.4) | 2 |
| 4.3. Theory and empiry | 5.3. Theory and empiry (4.3) | 2,3 |
| 4.4. Cognitive extent | 5.4. Creativity and imagination (4.4) | 2 |
| 4.5. Interdiscplinr. and knowledge flow | 5.5. Knowledge flows (4.5) | 1 |
| 4.6. Disciplinary strct. in country level | 5.6. Social and institutional (4.6) | 4,5 |

*5.1. Disciplinary Structure and Knowledge Flows*

The domain specificity and domain generality of NOS is one of the fault lines discussed already at length at the beginning of NOS movement. As Abd-El-Khalick [1] has argued, the question is largely about the level of sophistication and viability of in-depth discussion at different levels of education: simplified views needed at the school level but increasing sophistication in higher education and science teacher education. Here, we can sidestep problematizing the choice between generality versus domain specificity, and ask how the disciplinary structure of sciences might appear to us if we focused on how knowledge forms disciplinary clusters and how such knowledge flows between the clusters. Finding

out how disciplinary landscapes folds out in such an analysis provides a starting point to discuss how such structures may emerge and what processes might drive both the consolidation and the merging of disciplines. The analyses and findings of the disciplinary structure of sciences as based on SoS (Section 4.1) show that, while the disciplines are to certain degree isolated, there is merging of disciplines and formation of new disciplines on the boundaries of the merging disciplines. Between disciplines, one can find flows of knowledge between the boundaries, as well as within substructures within disciplines. This clearly indicates that, while there is a need to understand the nature of knowledge within disciplines, it is also important to understand the features of knowledge that allow flows of knowledge between disciplinary boundaries. Such features, if not entirely domain-general, need to be general enough to allow the crossing of boundaries on the large scale.

The results based on SoS do not reveal what actually happens at the boundary crossing and what kind of knowledge drives the flows. However, HPSS and PoS analyses and expertise suggest looking at theoretical structures and experimental methods, investigation and reasoning practices that may play key roles in shaping the disciplinary dynamics [107]. From this viewpoint, we can focus attention on how modelling practices and templates of models as parts of these practices cross the boundaries (for example, from physics, mathematics and computer science to fields of biology, economical sciences and even sociology) and templates originating from one branch become adapted and applied in new branches. Such boundary crossings shape the methodological practices and there is certain reciprocity; methodological practices provide identity to disciplines and shape formation of disciplines, but most practices are also born and developed within the disciplines. Similar notions can be made regarding experimental and methodological practices, for example from physics and electrical engineering, computer sciences to medical sciences and biomedical sciences. The number of examples is vast, and cartographic maps of the landscape of sciences provide a rich source of material.

### 5.2. Durability and Change of Scientific Knowledge

Examples of durability and change of knowledge in contemporary science are available from SoS in some specialised areas of knowledge (e.g., physics). Such discussions are illuminating even if more limited scope than studies within HPSS, which discusses long historical periods and cover different sciences. Examples of durability and change of scientific knowledge as based on HPSS scholarship thus remain invaluable for purposes of science education. Such views cannot be replaced by the findings provided by SoS, but together they may reveal a more complete picture, based not only on history of science but also providing perspectives to contemporary science, of how knowledge evolves and becomes accepted and established. The results of SoS about the subdisciplinary structure of physics and its evolution over the last hundred years shows how the roles of subdisciplines have waxed and waned, borders have shifted and new disciplines have emerged. Science and scientific knowledge are, from this viewpoint, evolving systems, where nothing remains immutable but evolves; new contributions to science produce siblings and gradually become distant ancestors, eventually fading away. In favourable cases (like many branches of classical physics), scientific contributions have long lasting impacts, and features of the initial contributions can be recognized and identified in many new generations. In insular and highly specialized subdisciplinary areas of physics, the lifespan of knowledge, as it can be identified as unchanged or based on identifiable contributions, is shorter. This, however, does not mean that such pieces of knowledge are lost; rather, they have more probably gained new life as part of newer advances. With regard to being identifiable as a durable part of the body of scientific knowledge, classical physics and interdisciplinary physics seem to have certain advantages over modern physics. However, several core contributions of lasting value can be identified in modern physics too.

Thus, far, SoS has focused on identifiable pieces of knowledge through citation analysis and concept-relatedness analysis of publications, and not so much through compiled and consolidated corpora of scientific knowledge, as exemplified by, e.g., monographs,

handbooks and textbooks. This limits the ability of SoS to draw conclusions about how the significance and durability of scientific knowledge evolves in time windows of centuries. However, the temporal evolution of pieces of knowledge already reveals much of the basic dynamics that eventually lead to the stabilization or extinction of scientific knowledge. Moreover, the detailed analyses based on SoS of a hundred years of physics [67], in the era when modern physics became established, provides interesting grounds to discuss the progress of science, either through evolution or more revolutionary instances. SoS rather supporting the former than latter position, although leaving plenty of room for both.

### 5.3. Theory and Empiry: Confluent or Convergent?

One of the topics discussed in textbooks of science, and especially of physics, is the relation of theory to empirical aspects (empiry) of science. A commonly held view is that theory and empiry are somewhat separate branches, not often interacting and very seldom merging in scientific research. However, as discussed in Section 4.3, this view has been challenged by the results of SoS. In this context, it is also fruitful to discuss, how scientific research is reported: is theory used by empirical research only as a motivational background, without really guiding research, or does it play a more fundamental role in experimental research? Evidence based on SoS as outlined in Section 4.3. May help to form a picture of science in which theory and empiry are seen in a more balanced way: construction of scientific knowledge driven equally by theory and empiry, borders between them being sometimes fluid. In physics, theory and empiry are indeed confluent and tightly interwoven (see Section 4.2). In biology, the situation is more complex; researchers think that there is little integration and that more is needed, while SoS suggests that the integration runs deeper than researchers believe. These topics and how SoS provides material for discussing them seem to fit well into the categories that Abd-El-Khalick [1] calls theory-laden NOS and empirical NOS.

### 5.4. Creativity, Imagination and Cognitive Content

That creativity and imagination are needed in scientific research may not be obvious at the school level, in particular if a conception of a recipe-like scientific method is taken for granted. At higher levels, however, such views appear to be obvious statements that provide no further insight on how imagination and creativity affect scientific research. Again, HPSS has provided interesting and insightful case studies of highly creative scientists and their contribution on science in history of science. One source that illuminates the role of creativity in contemporary science is found in SoS studies, about the cognitive content of science. The exploration of cognitive content provides information of how novel intellectual contributions appear and in addition, how the introduction of new concepts and ideas correlates with the number of researchers working together (see Section 4.4). These studies show that individual scientists have a significant role in feeding new concepts, and also that small groups of fewer than five scientists are highly productive in suggesting new concepts (and, apparently, new models or adjustments on theory). This provides a good starting point to discuss what happens to all these new inventions—whether they simply disappear, are somehow collated, or just found to be irrelevant. In addition, the studies suggest that the larger the group, more parsimonious the cognitive content of its work. This leads quite naturally to ideas and discussions to explore the role of large research groups in testing and validating knowledge, with the methodological and technological resources they possess. Such discussions have interesting extensions in the division of labour in science (related to the social-institutional factors of NOS and as they are discussed in FRA) as well as in the construction of new theories and their validation and testing.

### 5.5. Knowledge Flows between Sciences and Technology

Regarding the societal role of sciences, one key theme is how sciences, technology, and technological industries interact, and whether such knowledge flows only from science to technology or in both directions. Studies focusing on role of science in technology are at the

core of science and technology studies (STS) and in this, SoS comes very close to the goals of STS in its attempt to reveal and explore connections between science and technology. In Section 4.5, the knowledge flows between different sciences and between sciences, engineering and technology were discussed. SoS provides many interesting findings on such connections, pointing out the bi-directionality between science and technology. SoS provides strong support for the emergence of technoscience, where technology and science become inseparable and it becomes difficult (and perhaps unnecessary) to make a distinction between them. In big science, it is also common to find transaction zones [107], where science and technology are integrated so that the development of technology need by science is itself part of science, and new and important research problems advance scientific understanding through the development process. Such examples, which are discussed in SoS research but also in STS studies, are invaluable addition to NOS themes, easily finding their place in that framework, either in consensus NOS or FRA.

### 5.6. Social and Institutional Aspects

The social and institutional aspects of science finds clear actualizations in regional differences in how science is produced and consumed, and how different disciplinary areas are supported in different countries. SoS research has pointed out differences in how hard natural science, physics, and chemistry are more dominant in certain countries, due to their historical and ideological backgrounds, and how sciences supporting industrialization are supported and encouraged in countries seeking economic growth or improvement in technological competition. The correlations between the ideological and political history of a country, its structure of technology, and the focus of its scientific research are a rich source for discussions related to the values and aims of science and how larger scale societal factors affect such values.

Research in SoS has also revealed interesting changes in national distributions of institutions that produce scientific knowledge and that consume it. Some striking examples can be found from developments in European Union and North America (see Section 4.6). Results of SoS point out how the national policies and funding decisions guide research activities, affecting research careers and eventually educational structures and systems. On a local scale, the career paths of researchers reveal a strong dependence on institutional structure and national funding decisions, as shown by detailed studies of physicists' education and employment and their migration from research area to another, as discussed in Section 4.7.

Finally, the accretion of fame in science is one of the most well-known outcomes of the social dynamics within science. As many SoS studies show, there is strong accumulation of fame, leading to recognition of scientists and research contributions that significantly outstrip those of most other scientists and their contributions. Often, of course, such effect is genuinely due to the significance and novelty of the work done, but there is also strong component of fame attracting fame, often also called the Matthew effect [34]. The reasons and outcomes of such an effect, clearly visible in physics (see Section 4.2), provide a starting point for interesting discussions on how personal ambitions to gain fame and recognition are driving forces to do science, and how they might affect the science.

### 5.7. Possibilities and Options that SoS Opens Up for NOS

The summary of themes in this section (and as discussed in SoS in Section 4) are only a fraction of all the NOS-related items to be found in the fast-expanding SoS literature. The summary repeats the themes discussed in Section 4 with a purpose to show that that topics contained in SoS can be discussed in a framework of consensus NOS themes, as outlined for example Abd-el-Khalick [1], or alternatively, within categorizations provided by FRA. In particular, the way FRA sees role of disciplinary-structure and social-institutional dimensions seems to offer good ground for focused discussions paying attention on disciplinary differences, interdisciplinarity and boundary crossings. Similarly, FRA also provides good scaffoldings to discuss various topics related to social and institutional dimensions. How-

ever, although the starting points of consensus NOS and FRA differ, it is difficult to see insurmountable obstacles to fitting the SoS themes into the categorization within consensus NOS or FRA. This indifference to specific NOS schemes is an important notion because the evidence provided by SoS is based on scientometrics, big-data analysis, and data-mining, not on interpretative analyses based on preferred philosophical viewpoints. As such, it yields to discussions on several alternative viewpoints emerging from HPSS and helps to show that the goal is not to argue for a single correct position but to understand the differences between different positions and to appreciate the multifaceted and rich picture of science they provide. That picture may not be always coherent (no more than practicing scientists' views) but it might be interesting and inspire students' curiosity.

## 6. Implications for Science Teacher Education

The practical reasons to discuss SoS focusing on topics outlined in Section 4 is to make the science teachers familiar with sources of information contained in SoS and to make them aware what current research in SoS can provide for discussion of features of scientific knowledge, science practices and outputs of scientific research. In that, SoS may also provide material for NOS approaches and discussions of NOS. Similarly, SoS helps to embody the notion of social aspects of science through examples: how scientific communities act, what institutional and social organizations exist and how they operate, how and why scientists meet and discuss, how laboratories are run, what national differences exist and for what reasons, how national policies and agendas affect research, and so on. SoS provides vast amounts of material on all such topics. In addition, instead of using historical examples, as they are available from HPSS, SoS provides a source of contemporary examples. In short, science educators interested in developing NOS and using it as part of their teaching could benefit if they familiarized themselves with SoS to widen the perspectives of NOS.

The context in which SoS was discussed was part (three weeks) of a seven-week course (4 h per week + homework) for physics, chemistry and mathematics teachers in fall semester 2019 in a Finnish University (Faculty of Science). The part of that course involving SoS introduced topics as outlined in Section 4 and reflective discussions were conducted after the introductory lectures in each case. Each week, an article about SoS was provided for study, with a set of questions about the results and views presented in the article. The articles read and discussed were: Battiston et al. 2019 [66], Börner and Scharnhorst 2009 [49], Börner et al. 2006 [83], de Arruda et al. 2018 [68], De Domenico et al. 2016 [36], Herrera et al. 2010 [69], Leydesdorff et al. 2013 [39], Mazloumian et al. 2013 [89], Milojevic 2015, 2014 [70,80], Moya-Anegon and Herrero-Solana 2013 [55], Sinatra et al. 2015 [67], Wu and Wang 2019 [84]. These articles were chosen because they cover the topics discussed in Section 4 broadly enough, without being too specialized research articles. Moreover, these articles make a close contact with physics, chemistry and mathematics, areas which were of the most interest for pre-service science teachers in a faculty of Science. Therefore, the articles chosen were suitable for a course material by their scope, length and depth. The articles by De Domenico et al. [36] and Sinatra et al. [67] were given more attention than the other ones. Students were not asked to familiarize themselves in detail with all material in articles, only parts they found of interest after perusal. The homeworks contained five questions, of which four were simply focused on articles and framed so that they guided attention on topics as outlined in Section 4. In each homework, one question for students was what was new for them and what they found to be the most interesting topics. The feedback reported here is based on that question. The different educational views of NOS were not discussed during the introduction. However, thematic topics as they appear in consensus view of NOS were introduced (but avoiding the special terminology of consensus NOS) and discussed after the three-week period focusing on SoS. In reflection and discussion, however, the facts brought forward by SoS were discussed from the point of view of how they might be related to knowledge and its epistemology in general, as well as the conceptualization and progress of science. Students were free to

discuss the epistemological or science philosophy viewpoints they were familiar with (e.g., inductive and hypothetic deductive positions, realistic or constructivist positions). The lecturer also brought in viewpoints that augmented the students' views or put them in more specific contexts (e.g., versions of realism, constructive or critical, constructive empiricism and different views of the conception of scientific progress etc.). Such additions to the discussion were not planned or predetermined; instead, they were students' spontaneous responses to the directions that the discussion took in reflections. No systematic record was made of these instances.

The topics that were new and interesting for students reveal that, even in an instructional setting that is neutral with respect to philosophical stances and that does not explicitly introduce any tenets or positions for students to adopt with regard to science, one can find that SoS topics guide attention to similar kinds of themes and conclusions that are the backbone of NOS. In what follows, several responses that parallel NOS views are reported. The following is a list of examples of students responses (translated from Finnish).

1. The disciplinary and interdisciplinary structure of sciences.

- I had never before come to think about how much research is being done in science today and how the number of research outputs is growing at such a tremendous rate. Exponential growth was mentioned both in the article and in the lecture.
- I was surprised by how large is the [difference] in the number of studies published in medicine in comparison to physics and chemistry, which I thought would be one of the greatest fields of research.
- The lectures and the article provided a lot of new information about the sizes of different disciplines. I had not thought biomedicine to be so great that it could not be "viewed on the same map" as, for example, mathematics, physics, or the social sciences. [It is] intriguing how the visibility of such a large discipline in the school world is so low.
- The lectures and the literature broadened my understanding ... of sciences, the connections between them, the different "profiles" of scientists, and the whole field of science and the "scope" of different disciplines.
- Before I had not realized that the disciplines are so interlinked as they really are. Significant results in physics are [also] published in non-physics series. Perhaps [we need] to redefine physics. I am wondering whether it is necessary to distinguish (or maintain historical distinctions) between different natural sciences, which, however, describe the same world? [M]ultidisciplinarity has been explicitly sought in modern society. On the other hand, interfaces between disciplines are a natural consequence of the advancement of each discipline.
- An interesting, new thing [was] that research in the field of natural sciences has become more multidisciplinary and focuses on softer themes (life sciences). I welcome this development and believe that with more diverse interdisciplinary co-operation, more significant findings can be made through science.

2. The cognitive content of studies, introduction of new concepts.

- The dependence between the size of the research group and the publication of new concepts [as] presented in the lecture was new. However, it was perfectly reasonable how in physics the curve [of the dependence] was as shown.
- Although publication volumes are growing exponentially, cognitive content is growing linearly, that is interesting. It supports my own observation that the average "additional information" produced by one article has been declining steadily. On the one hand, this makes it almost impossible to follow the new literature, on the other hand, it gives hope that it will possible to keep up with new knowledge.
- The lectures revealed that groups in small scientific communities solve and produce more new [concepts] compared to larger groups. It was interesting to get to see the charts [showing] this.

- The information that new scientific insights often take place among individuals or small research groups is an interesting detail.

3. Cognitive content, theory construction and testing.

- I was also interested in theories of the formation of scientific knowledge. Again, the variety of theories [of formation of scientific knowledge] is surprising. So I would like to know more about the philosophy of doing science, and what, according to different views, is the goal of science.
- It was interesting [to note] that concepts are not eternal, they just have a life cycle and a purpose and eventually they die. Concepts can change over the life cycle.
- I was interested in [the notion that] the same concept can mean different things to different people. When does a concept turn into something different, and when does only its content live in time? Who is entitled to decide? In science, different perceptions arise (hopefully) mainly from different starting points and perspectives.
- Scientific concepts have always been unchanging things for me. Something that has been agreed upon for what it is and then it is just that. It was new to me, therefore, to become aware of ... the evolution of scientific concepts; they expand and converge, their connections to each other change, their purpose evolves, or they can take on a whole new form of presentation. When scientific progress has taken place, developments in concepts have always caused a chain reaction that changes all the concepts associated with the original. Fabulous!

4. Social aspects, scientific activities and scientific communities.

- The social side of science and the examination of the scientific community as part of science aroused my interest and reflection on how strongly science and the scientific community are linked. Can science exist without a community structure?
- I liked a lot [to learn] about the importance of different networking [of scientists], and how to find key people who link [disciplinary] areas to each other.
- The most important aspect is definitely that there is no single answer to the nature of scientific knowledge, but it is a combination of features that depends on both the field of science and the scientific community. Science is also influenced by history, and with the present, it gives direction to the future work of science.

5. Production and consumption of scientific knowledge and role of national policies

- It was interesting to see on which different continents and in which states the creation of scientific knowledge was focused.
- Maps and models of where scientific information is produced... brought a huge increase in perspective to my own perceptions of, for example, how large certain concentrations of scientific knowledge are compared to others.
- I found the various diagrams describing physics as a global phenomenon to be of particular interest — especially what kinds of physics has been studied in different countries. The network of connections between different disciplines also sparked ideas, and it was fun to see how much the different disciplines touch and interact with each other.

6. Other general notions of interest here included.

- Other courses talk a little about measuring the effectiveness of science. This [is] perhaps familiar for researchers through their own academic careers, but for us [pre-service] teacher students, it is interesting information that might otherwise go unnoticed. In the same way, a teacher may not have much knowledge about the big picture and interaction of sciences (networks of sciences). It was interesting to see diagrams of what places and in what research groups science is done.
- I wasn't [aware] of science that explores science itself and that science can be studied, and that it can be so interesting. The new thing was also how difficult it is to define a

researcher's contribution to scientific work. I was [familiar with] studies of the growth of science at different times, but not from this point of view and so widely.

- It was interesting to see how even in scientific research, it is not clear where a researcher will end up and what kind of a research field in their or her career. Research areas within physics, for example, can change radically during the career path. I had assumed that a researcher's career orientation would already be fixed at the training stage. Obviously, this is not often the case.
- During the lectures, my conception strengthened, that science cannot be given an exact definition, but that its nature includes diversity and a certain kind of fragmentation. However, a considerable number of features that are characteristic of science can been identified. I find it interesting how we can identify the features of science and describe what science is like, but still we cannot give a proper definition of science.

The excerpts from the students' responses show that introducing SoS, with minimal additional philosophical interpretations and discussions, but simply as evidential facts and findings, sparks ideas and views that come very close to NOS themes and tenets, but now through spontaneous reflection. They thus offer students a starting point for deeper discussions and considerations how the big picture of science or a more panoramic view builds up if topics are approached from some variety of NOS views, or from perspective of some preferred philosophical stance, e.g., some variety of realistic positions, constructive empiricism or constructivism. Here, and in the course, no preference was made between such views and positions, either related to NOS or philosophy of science. Different views were treated as equally possible and tenable positions, based on chosen emphasis on aspects of interest, which can support the needed flexibility for different rationalizations and different argumentations behind various possible views on science.

Based on the responses to the question of what was new and most interesting in the topics discussed, it appears that the students' factual knowledge of science was weak with regard to many topics: disciplinary structures, how the sciences produce and consume knowledge, and how scientific institutions operate and interact with society. The introduction to SoS and its results may then provide a concrete grounding for more philosophically-oriented discussions, without specifying any particular philosophy of science (constructivistic or realistic or any other) as the preferred and desired view, but to use different views as lenses to reveal different aspects of science and make them conceivable through the given viewpoint. There is then no need to try to find an essence of science, nor closed sets of characteristic features, nor aims and values of science that exist independently beyond the aims and values of scientists, nor norms that are given and which scientists follow, rather than being agreed by scientist and their communities.

## 7. Discussion

The picture of sciences, its disciplinary structure, substructures, knowledge flows and its internal sociodynamics, as revealed by SoS, is based on textual or lexical analysis of scientific documents. SoS treats such documents as trails to be followed in exploring the evolutionary paths and landscapes of science; it is a cartography of science. Such an approach has the advantage of being evidence-based, obtaining its understanding from analysis of data and records. For science education, a combination of SoS with established approaches in NOS (like consensus NOS and FRA) provides a valuable way to enrich our understand of NOS from complementary perspectives. Results and findings based on SoS are also important ways to extend the HPSS-based views on science in a direction that is more attentive to the large-scale structures of contemporary sciences. NOS provides different lenses or windows to interpret the evidence-based picture of science provided by SoS. While the data that SoS provides is composed of fragments and piecemeal patterns of science based on its practices, activities and products, NOS can provide viewpoints and approaches to construct personal and panoramic pictures, worldviews where the fragments gain value and significance in building a personal picture of science, in accordance with personal convictions and wider worldviews. Such broader perspectives may well be rooted



in certain rationalistic philosophical stances, be they some variety of realism or radical constructivism, materialism or idealism. The goal of NOS is then not to select a correct or best view, or a view best conforming to some philosophical position or the positions of practising scientists, but to evaluate and argue for the chosen view and be able to discuss how the evidence that we have of science supports a particular view.

Interestingly, with regard to practical teaching, the discussion above comes close to the problematics encountered in teaching history. If SoS provides the facts to be discussed and evaluated, and NOS the meta-level viewpoints, a question arises as to which comes first. In teaching history, one is encountered with the situation that, before a deeper understanding is possible, a vast amount of detail must be known and acquired. This is a well-known challenge in learning general history [108,109]. Only when a rich enough knowledge base becomes available does it become possible to start to construct a big picture, a landscape of history, and science history as part of that history. From this vantage point, different versions of NOS naturally provide conceptual scaffoldings, which can then be embodied and enfleshed by the facts and details provided by SoS. The role of the scaffoldings is thus not to provide established, "correct" interpretation of the phenomena, whether history or NOS, but to provide tools to construct arguments supporting one's positions and provide meaning and purpose for the facts so that they can be used to build a personal worldview and set of values.

The feedback received from a teacher education course in which SoS was introduced shows that many views paralleling NOS themes emerge quite spontaneously, even when no attempt was made to use NOS scaffolding as guidance for discussion. This strongly supports the view that NOS would provide a quite natural and viable scaffolding, which is useful and meaningful for guiding and focusing discussion of SoS findings and results. NOS, equally well in form of consensus NOS or FRA, would align quite well with views arising from SoS. Here, however, the course which is reported attempted not to establish explicit connection to NOS. It was, however, important to carry out the pilot version without explicitly using the NOS schemes in order to see whether SoS would resonate spontaneously with such ideas. Now, when the feedback supports this, the next step planned is to develop a course in which NOS issues more explicitly discussed, but only after the SoS findings and results have been introduced. It might be that such a combination of SoS and NOS does not entirely follow how NOS was meant to be used. However, at least for science teacher education, such approach has the advantage of relieving the tensions between NOS and students' initial expectations of how science can be approached through the lenses of NOS.

Finally, it seems that an approach using SoS is also quite viable for school-level teaching. The introduction of the topics discussed in SoS are not so different or so much more complicated than those encountered in history, for example. In teaching and learning history, it has also long been recommended that the it is first necessary to learn facts, temporal ordering and thematizations (here, SoS), and only after that, the interpretations that provide a deeper understanding and a panoramic view of the history (here, NOS-like interpretations). In many ways, history and societal studies are quite natural companions to NOS and better discourse between these school disciplines may benefit both. Both have the tension between factual contents, a vast number of interesting and important details, the kaleidoscopic tendency of facts to arrange in various constellations according to the different viewpoints chosen, and the difficulty of providing meta-level panoramic views, requiring sufficient knowledge about the factual content before a meaningful big picture can be formed.

**Funding:** This research received no external funding.

**Informed Consent Statement:** Informed consent was obtained from all students involved in the study. According to regulations of country where study was carried out, approval from an ethical board is not required.

**Data Availability Statement:** Data are contained within the article.

**Conflicts of Interest:** The author declare no conflict of interest.

**Abbreviations**

The following abbreviations are used in this manuscript:

SoS     Science of Science
NOS     Nature of Science
FRA     Family Resemblance Approach
PoS     Philosophy of Science
HPS     History and Philosophy of Science
HPSS    History, Philosophy and Sociology of Science
STS     Science and Technology Studies
PSP     Philosophy of Science Practice

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
