# Peer review of "Nature of Science (NOS) Being Acquainted with Science of Science (SoS): Providing a Panoramic Picture of Sciences to Embody NOS for Pre-Service Teachers"

_education, doi:10.3390/educsci11030107_

Round 1

Reviewer 1 Report

Excellent research

Author Response

I thank all reviewers for their valuable comments and I am very happy to find out that they have found the topic of interest. I have tried my best to follow the advice and comments, and to respond to criticism. In some few cases I have found it difficult to follow the suggestion, I have tried to justify my point and rationalization behind it.

I have tried to find and correct all typos and misspellings as carefully as possible (and, sorry to say, found still all too many). If it seems that the text needs language editing, I am willing to arrange such to be done, but that will probably take about two to three weeks.

In attached file you will find my detailed comments. The locations of changes are highlighted in the manuscript.

Reviewer 2 Report

In this paper trying to connect SoS with NOS, there are some issues to be afforded:

-In the abstract, HPSS appears without previous introduction, as conversely it is done for SOS or NOS.

- It would be helpful to find in section 5 a table/figure including the contributions to NoS from SoS. I assume they are the headings of the subsections, but it would be clearer in this other format.

-Regarding section 6, the use for teacher training of a proposal including SoS, again to include it not only in text that is hardly followed, but also in a scheme in a figure showing the sequence, would help the reader guess how it was implemented.

Others: Regarding in text references, some parenthesis are not closed: see line 44, 58, 91, and some more through the text.

Author Response

I thank all reviewers for their valuable comments and I am very happy to find out that they have found the topic of interest. I have tried my best to follow the advices and comments, and to respond to criticism. In some few cases I have found it difficult to follow the suggestion, I have tried to justify my point and rationalization behind it.

I have tried to find and correct all typos and misspellings as carefully as possible (and, sorry to say, found still all too many). If it seems that the text needs language editing, I am willing to arrange such to be done, but that will probably take about two to three weeks.

In an attached file you will find my detailed comments. The locations of changes are highlighted in the manuscript.

Reviewer 3 Report

This manuscript presents several examples of how Science of Science (SoS) can contribute to a better understanding of science, providing new approaches to develop understandings about Nature of Science. In spite of the relevance and originality of the manuscript, there are some aspects that have to be improved:

  1. I don't feel qualified to judge about the English language and style. However, it seems that the manuscript should be revised regarding English language. For example, on line 33 “such discussion .. are/is”; on line 164 “five scholars … was/were”; lines 482 – 483…

  1. Also, some mistakes on line 17; and lines 109 and 775, “NoS or NOS?”; on lines 144- 146 (What do you mean by PSP and SPS?). On line 389, “Milojevic 2015” should be replaced by number 70. See also line 778, 852…

  1. Concerning the structure of the manuscript:

a. The abstract is confusing, as it is mentioned a science teaching course and students. It is not clear if those students are preservice teachers or not.

b. The consensus view of NOS and FRA are mentioned in the introduction, as well as throughout the manuscript. However, those approaches must be clarified and better described. For example, there are some references to “the theory-ladenness of science” throughout the manuscript. However, this concept is not described and clarified in the introduction.

c. On line 183, it is not clear which recent study is that.

d. Theoretical framework is too long. It should be more concise. It is even referred by the authors, on lines 276-277 “Further discussion of that possibility, however, is beyond the scope of the present study.”.

e. Concerning methodological aspects, why were those articles chosen to the teachers’ course (lines 1048 -1051)?

f. Results should be presented in an organized way, instead of transcribing students’ responses.

Author Response

(The authors gave the same response as above.)

Round 2

Reviewer 3 Report

I know that the terms are quite familiar, but I consider it important to specify the terms that we are using. I understand and accept your viewpoints.

There are some parenthesis that are still not closed (for example lines 34, 42 e 43). Check the "to to" on line 889.

Author Response

Thank you pointing out some final adjustments that needed to be done.

I have added the definitions of theory-ladenness and social-embeddedness (lines 127-131), immediately after the lines where they are introduced as part of NOS. I think these were the terms the reviewer was asking to be defined better.

In addition, the remaining typos as pointed out by the reviewer are now corrected.

I hope that now all necessary corrections are completed adequately.